# Influence of Potentiostat Hardware on Electrochemical Measurements

**DOI:** 10.3390/s24154907

**Published:** 2024-07-29

**Authors:** Abhilash Krishnamurthy, Kristina Žagar Soderžnik

**Affiliations:** 1Department for Nanostructured Materials, Jožef Stefan Institute, Jamova 39, 1000 Ljubljana, Slovenia; kristina.zagar@ijs.si; 2Jožef Stefan International Postgraduate School, Jamova 39, 1000 Ljubljana, Slovenia

**Keywords:** electrochemical sensing, potentiostat, hardware, analogue, digital, cyclic voltammetry

## Abstract

We describe two operating modes for the same potentiostat, where the redox processes of hydroquinone in a hydrochloric acid medium are contrasted for cyclic voltammetry (CV) as functions of a digital/staircase scan and an analogue/linear scan. Although superficially there is not much to separate the two modes of operation as an end user, differences can be seen in the voltammograms while switching between the digital and analogue modes. The effects of quantization clearly have some impact on the measurements, with the outputs between the two modes being a function of the equivalent-circuit model of the electrochemical system under investigation. Increasing scan rates when using both modes produces higher peak redox currents, with the differences between the analogue and digital modes of operation being consistent as a function of the scan rate. Differences between the CV loops between the analogue and digital modes show key differences at certain points along the scans, which can be attributed to the nature of the electrolyte affecting the charging and discharging processes and consequently changing the peak currents of the redox processes. The faradaic processes were shown to be independent of the scan rates. Simulations of the equivalent-circuit behaviour show differences in the responses to different input signals, i.e., the step and ramp responses of the system. Both the voltage and current steps and ramp responses showed the time-domain behaviour of distinct elements of the equivalent electrochemical circuit model as an approximation of the applied digital and analogue CV input signals. Ultimately, it was concluded that similar parameters between the two modes of operation available with the potentiostat would lead to different output voltammograms and, despite advances in technology, digital systems can never fully emulate a true analogue system for electrochemical applications. These observations showcase the value of having hardware capable of true analogue characteristics over digital systems.

## 1. Introduction

Utilizing the available hardware to its maximum benefit also extends to phenomena that can sometimes go unnoticed to most end users. While modern devices with their well-optimized, cutting-edge “digital” hardware [1] and software advances can obtain the desired results for most measurements, they are still only approximating the behaviour of a true “analogue” device [2]. In order to fully understand the differences between digital and analogue devices, a deeper look is necessary into how the data signals [3] are transmitted in computation. Analogue systems are continuous in nature, while digital signals are discrete. In particular, the digital signals used in computing are predominantly encoded as binaries made up of 0s and 1s. The modern digital computing [4] revolution has rendered older analogue computing systems [5] almost obsolete, as digital devices possess two major advantages over their analogue counterparts. Firstly, digital systems allow the use of memory devices based on semiconductors [6], which can store the information more reliably compared to analogue memory devices such as magnetic tapes [7]. The second advantage is that digital signals and systems are inherently more immune to noise [8] when compared to analogue ones since the signals are encoded as binary [9]. However, this encoding also brings with it some drawbacks, such as a time lag since most of the signals originating in nature are analogue and would need to be encoded into a digital signal to be read by a digital computer. This encoded signal is a mere approximation of reality. Since the data must be made to fit into discrete chunks, quantization errors can lead to compounding, especially with numbers that have extremely large or small magnitudes. This is especially noticeable in, for example, sensors, which generally operate on analogue inputs. These data lose their integrity during the encoding and decoding processes [10]. Figure 1 shows the difference between an analogue and a digital signal, albeit exaggerated as these differences are usually only visible upon close inspection with special hardware. This is because modern devices now have the ability to produce nearly analogue signals using digital hardware.

The use of analogue [11] hardware in an electrochemical [12] potentiostat [13] offers an optimistic change towards a better understanding of the processes that occur when subjecting electrochemical systems to traditional measurement techniques [14]. Focusing on one method, cyclic voltammetry (CV) [15], the Metrohm VIONIC potentiostat [16] offers the user the choice of performing CVs as either digital (staircase) (Figure 1b) or analogue (linear) (Figure 1a). The CV is generally accomplished by starting the measurement at a given potential, increasing or decreasing the potential to another value over a given period at a predetermined rate of change in the potential called the “scan rate” and then returning to the initial potential, hence, cyclic. The rate of change in the potential plays a large role in electrochemical systems as the electrolyte is inevitably subjected to the charging and discharging processes of the double layer [17]. The double layer is modelled as an imperfect capacitor and represents the non-faradaic processes, whereas the redox processes are represented by a resistor and correspond to the faradaic processes. Most electrochemical systems also consist of an uncompensated resistance in series, usually the resistance of the electrolytic solution; these three components are shown in Figure 2, the simplest electrical model of an electrochemical system known as a Randles’ circuit [18].

In order to understand the charging and discharging behaviour of a capacitor, let us take a look at the circuit in Figure 3, where a resistor and a capacitor are connected in series to a power source and the output voltage is the voltage across the capacitor.

The electrical circuit forms a first-order system [19] since the capacitor is the only energy-storage device. Solving the first-order differential equation gives the general solutions in Equations (1) and (2).

For charging,
(1)$$V_{O}=V_{i}(1-e^{\frac{-t}{RC}})$$

For discharging,
(2)$$V_{O}=V_{i}(e^{\frac{-t}{RC}})$$

Here, *V_O_* is the output voltage (across the capacitor), *V_i_* is the input voltage, *R* is the resistance, and *C* is the capacitance. The factor “*RC*” is defined as the time constant of the circuit, denoted by τ. When *t* = τ, we obtain *V_O_* = 0.632 *V_i_* for charging and *V_O_* = 0.368 *V_i_* for discharging. Hence, the definition of the time constant is the time required for the output voltage to rise to 63.2% of the input voltage during charging or fall to 36.8% of the input voltage during discharging. As long as the *R* and *C* values are fixed, the time constant is also fixed. If *t* = 5τ, *V_O_* = 0.993 *V_i_* ≈ *V_i_* for charging and *V_O_* = 0.006 *V_i_* ≈ 0 for discharging. Thus, it takes about five time-constant durations for the output to reach nearly the same value as the input. Figure 4 shows the charging and discharging curves of the circuit in Figure 3, indicating the time constant for both scenarios.

Looking at Figure 1 again, which shows that analogue signals are smooth and continuous while digital signals are discrete with abrupt jumps, each abrupt jump can be considered as a voltage step, either up or down, as shown in Figure 4a,b. Given that the Randles’ circuit shown in Figure 3 has a capacitor, the electrochemical system will be subjected to charging and discharging with every quantization jump of a digital signal and thereby experience a comparative “lag” in contrast to the smooth analogue curve. For the end user, this is to be verified with experimental data comparing the analogue and digital CVs.

The state-of-the-art electrochemical potentiostats on the market are still predominantly digital-hardware-based systems, subject to all the aforementioned pros and cons inherent to digital electronic systems. Electrochemical potentiostats with analogue-hardware-based systems have existed in a niche space where the accuracy of the measurements has priority over the device complexity and cost [20], usually as an application-specific add-on [21]. However, now with devices such as the Metrohm VIONIC including the analogue hardware as an option for end users, the prospects of having general-purpose analogue computers as a readily available measurement tool in electrochemical labs adds considerable value when studying electrochemical systems much more thoroughly.

## 2. Experimental Section

Materials: A 5 mM hydroquinone solution in 1 M HCl was prepared using analytical grade hydroquinone (>99%) and 37% HCl purchased from Sigma Aldrich, mixed with Milli-Q deionized water. Sensor elements were prepared using Metrohm DRP150 electrodes (working electrode: carbon; counter electrode: platinum; reference: silver) modified with solution of 20% platinum on carbon black (Vulcan XC-72R) catalyst purchased from Fuel Cell Store, 1 W Bronze Ln, Bryan, TX 77807, USA with Pt particle sizes of 2–3 nm. An ink of the aforementioned 20% Pt on carbon was prepared by dispersing 1 mg of the catalyst in a solution containing 0.995 mL of absolute ethanol and 5 μL of 5% Nafion solution in ethanol as the binder. Then, 10 μL of the catalyst ink was drop cast only on the working electrode after cleaning the surface with DI water and ethanol and dried on a hot plate at 60 °C. The sensor elements were then allowed to cool and dry overnight in a cool, dark environment before being used for measurements.

Instruments: CVs were measured using a Metrohm VIONIC, on loan from Primalab d.o.o., Pod Bregom 27, 3313 Breg pri Polzeli, manufactured by Metrohm AG, Ionenstrasse, 9100 Herisau, Switzerland Parameters for CV: scan window: 0 V to 1 V; scan rates: 10 mVs^−1^, 25 mVs^−1^, 50 mVs^−1^, 100 mVs^−1^; number of scans: 3; potential step size: approximately 1 mV (digital). All mentioned potentials were against a silver reference electrode. All electrochemical measurements were conducted in a climate-controlled laboratory setup at STP.

Simulations were performed with MATLAB R2017b using the control systems toolbox to obtain the step and ramp responses.

## 3. Results and Discussion

### 3.1. Effect of Scan Rate

The scan rate plays an important role in the value of the measured current in all the potentiodynamic electrochemical measurements. The simplest definition of the scan rate is the rate of change in the potential with respect to time. The instantaneous current flowing through a capacitor [22] is given by Equation (3).
(3)$$i_{C}=C_{dl}\frac{dV}{dt}$$
where $i_{C}$ = capacitive current (A),$\;C_{dl}$ = double-layer capacitance (F), and *V* = applied potential (V).

The capacitive current is directly proportional to $\frac{dV}{dt}$, which is the rate of change in voltage with respect to time, i.e., the scan rate. Faster scan rates should yield higher overall measured currents, simply due to the larger contribution of the capacitive current compared to the faradaic current. This is evident in Figure 5, comparing analogue and digital CVs at different scan rates.

In both the analogue and digital CVs, we can see that increasing the scan rates produces a higher overall measured output current as we go from 10 mVs^−1^ to 25 mVs^−1^, 50 mVs^−1,^ and 100 mVs^−1^ despite having the same concentration of hydroquinone (5 mM) [23]. This agrees with the theory that the increase in current is due to the increase in the capacitive current only, with no contribution from the faradaic current of the redox of hydroquinone.

### 3.2. Comparing Analogue and Digital Measurements

Although Figure 5a,b seem to suggest that the analogue and digital CVs are identical, there are some differences visible upon closer inspection of the curves. The main difference is that the currents produced with the digital CVs seem to be higher compared to the currents produced with the analogue CVs, comparing the Y-axes of both graphs. In order to take a deeper look, comparing each analogue CV to the digital CV measured at the same scan rate should provide a better perspective, as show in Figure 6.

Looking at each scan rate individually and comparing the analogue and digital measurements, subtle differences can be observed. While the overall profile of both CVs appears almost identical, in every measurement the digital measurement produces a higher peak current for both the oxidation of hydroquinone to 1,4-benzoquinone and during the reduction in 1,4-benzoquinone back to hydroquinone. There is also a subtle bump in the analogue curve where it momentarily produces a higher current compared to the digital CV between 0.5 and 0.6 V vs. Ag reference. Both these differences can be attributed to the nature of the CVs themselves: as the analogue CV is smooth, it produces a more natural output-current curve. The digital CV has a quantization limit, and being abrupt is probably subjecting the electrolytic solution of hydroquinone in hydrochloric acid to experience a “lag”, which would appear due to minor charging and discharging of the solution in an attempt to follow the sharp changes in the digital signal. This would explain why the higher peak currents are observed during both oxidation and reduction, as the electrolyte is not undergoing as much of the charging and discharging with the analogue signal due to the smoother and more gradual rise and fall in the potential over time.

While on the surface it is easy to associate all the noticeable changes to the switching of the operational modes on the potentiostat, the internal electronics and the employed algorithms within the device also play a key role in how the data obtained are processed. Quantization errors caused by the conversion of the data between analogue and digital form, which is necessary for the data to be processed by the computer, can also be a factor, and the total harmonic distortion of the analogue-to-digital converter (ADC) determines the integrity of the signal after conversion, which is a function of the resolution of the ADC. However, this information is not readily available to the end users and, for all intents and purposes, the potentiostat is a black box for the end user.

### 3.3. Simulation of Equivalent-Circuit Behaviour Using MATLAB

Assuming the electrochemical system to be ideal, the behaviour can be simulated by analysing the Randles’ circuit shown in Figure 2. The primary elements of interest are the polarization resistance, the path through which the current produces useful work, and the double-layer capacitance, which is responsible for the observed charging–discharging effects. By determining the transfer functions for the element of interest, the voltages across and the currents through all the circuit elements can be expressed as a function of the input parameter.

### 3.4. Step Response and Ramp Response

The step response and the ramp response of any system are defined as the system’s behaviour when subjected to the unit-step function and the unit-ramp function, which are shown in Figure 7 [24].

The unit-step function, also known as the Heaviside step function, is defined as follows in Equation (4).
(4)$$H(t)\;or\;u(t)=\{\begin{array}{c} 1,\;t\geq 0\\ 0,\;t<0 \end{array}$$

The unit -ramp function is defined as follows in Equation (5).
(5)$$r(t)=\{\begin{array}{c} t,\;t\geq 0\\ 0,\;t<0 \end{array}$$

Both the above functions are continuous. The digital counterparts of the unit-step and unit-ramp functions are *u*(*n*) and *r*(*n*), where n represents discrete sampling instants instead of a continuous time scale on the time axis, and, in the case of the ramp function, the values increment in fixed, discrete steps along the amplitude axis. The analogue waveform is a true-ramp function, whereas the digital waveform is approximated to be a ramp function by adding a series of digital-step functions, which can be mathematically described as in the following Equation (6).
(6)$$r(n)=\underset{\begin{array}{c} ∆t\to 0\\ ∆a\to 0 \end{array}}{\lim}k_{a}∆a\sum_{i=0}^{\infty }a_{i}u(n-k_{t}∆tn_{i})$$
where *k_a_* = amplitude scaling factor, ∆*a* = limit of amplitude resolution, *a_i_* = amplitude index, *k_t_* = time scaling factor, ∆*t* = limit of time resolution, *n_i_* = time index, and *u*(*n*) = digital unit-step function.

By looking at how the equivalent circuit responds to the voltage and current steps and ramp functions, we can gain an understanding of why there are differences between the analogue and digital inputs.

For ease of analysis, consider three different electrochemical systems with the following parameters. For system 1, series resistance R_S1_ = 100 Ω, polarization resistance R_P1_ = 25 Ω, and double-layer capacitance C_dl1_ = 1 μF. System 2 retains the same values for the series and polarization resistances: R_S2_ = 100 Ω and R_P2_ = 25 Ω but has a higher double-layer capacitance C_dl2_ = 10 μF. System 3 also retains the same values for the series and polarization resistances: R_S3_ = 100 Ω and R_P3_= 25 Ω but has a lower double-layer capacitance C_dl3_ = 0.1 μF. By calculating the transfer function for the voltage across the double-layer capacitance/polarization resistance and obtaining the time-constant R_S_C_dl_, increasing or decreasing C while having the same value of R has a corresponding effect on the time constant. So, system 2, having a value of C 10× larger than system 1, should have a time constant 10× larger, making it slower to respond, while system 3, having a value of C 10× smaller than system 1, should have a time constant 10× smaller in theory.

#### 3.4.1. Voltage Step and Ramp Response

Since the polarization resistance and double-layer capacitance are in parallel, the voltages across them are the same and, therefore, the voltage-step and ramp responses are identical for both. Figure 8 shows the voltage-step response across the two elements.

The black curve shows the unit-step function, while the red curve shows the voltage-step response for system 1. System 2, having a larger time constant (green), takes a longer time to reach the final value of 1 V compared to system 1, while system 3, having a smaller time constant, reaches the final value significantly faster than system 1, as expected.

Figure 9 shows the voltage-ramp response across the two elements.

While in the case of the step response, differences can be seen with respect to the different behaviour of the systems, for the ramp response, the curves lie on top of each other over a time scale of 0.5 s. However, when viewed on a shorter time scale, the differences start to become apparent, as shown in Figure 10.

Here we see, once again, the baseline (red) system is slightly lagging the input-ramp function (black). System 2 (green), with the larger time constant, is considerably slower than the baseline, while system 3 (blue), with the smaller time constant, is much quicker to follow the input voltage. However, once the time scale becomes large enough, these differences become less apparent and the curves overlap, making it appear as if there are no discernible differences between them.

#### 3.4.2. Current-Step Responses

Simulations were also performed using the control systems toolbox in MATLAB to obtain the current-step responses through R_p_ and C_dl_, as shown in Figure 11 and Figure 12.

The black curve shows the unit-step current function, while the red curve shows the step response of system 1 (baseline). System 2, which has a time constant 10× larger than system 1, shows a slower step response (green), as expected, taking considerably longer to reach the final value of 1 A, while system 3, having a time constant 10× smaller (blue) than system 1, reached the steady-state value much more quickly than system 1, which is once again in agreement with the expectations.

The current-step response through the double-layer capacitance shown in Figure 12 follows the same trend as the polarization resistance. The black curve shows the current rising to 1 A at 0 s and, in contrast to the current-step response of the polarization resistance, the currents through C_dl_ start at 1 and settle at 0 A. The baseline (red) is set by system 1, while system 2 (green), having a time constant 10× larger in magnitude, is considerably slower to reach the final value of 0 A, and system 3 (blue), with the smaller time constant, settles to 0 A much more quickly than system 1, in agreement with the assumptions.

The explanation for the current-step responses through R_p_ and C_dl_ makes sense, as the currents through R_p_ and C_dl_ add to the total current flowing through the network. So, as the current through one element increases, the current through the other should decrease as their sum cannot exceed the source current. The current rises through R_p_ because the voltage across is also rising. For C_dl_, it is slightly more complicated; because the voltage across the terminals is increasing with time, the capacitor charges and slowly starts to apply a voltage opposing the source, steadily decreasing the current through it, and, when fully charged, completely blocks the current.

#### 3.4.3. Current-Ramp Responses

The current-ramp responses through R_p_ and C_dl_ were also obtained with MATLAB simulations, using the control systems toolbox. The current-ramp response through the polarization resistance R_p_ is shown in Figure 13.

Considering a longer time scale (0.5 s), all three systems (red, green, and blue curves) generally follow the input current-ramp function (black curve) as it rises linearly with time, indistinguishable from each other. However, when viewing a much shorter time scale, as shown in Figure 14, the differences become much more apparent.

Once again, the baseline (red) system 1 is trying to follow the input-current ramp (black) as closely as possible, but a minor divergence can be seen. System 2 (green), with the larger time constant, shows a much larger deviation compared to the baseline, while system 3 (blue), with the smaller time constant, is noticeably quicker when following the input. These results logically follow the same pattern as the step responses, which confirms the theoretical assumptions with respect to the time constants of each individual system. However, once the timeframe becomes large enough, these minor deviations become insignificant and it appears as if the output follows the input without any lag, as shown in Figure 13.

The current-ramp response through C_dl_ is shown in Figure 15.

Looking at the final settling values of all the systems, they appear to be suppressed to 0 A, while the unit-ramp current function increases linearly with time; this makes sense as the capacitor will not be receiving any current, as the polarization resistance is essentially taking all the input current through its branch, as shown in Figure 13. Taking a closer look by zooming in on the time and current axes, Figure 16 shows some minor differences.

In this instance, minor variations to the behaviour of the systems can be observed, as once again system 1 (red) sets the baseline, which shows a minor increase in current before settling to a stable value of around 25 μA. At the same time, system 2 (green), with the larger time constant, reacts much more slowly as it rises to settle at almost 250 μA and is not suppressed as quickly as system 1. System 3 (blue), with the smaller time constant, is almost at 0 A, reacting very quickly and suppressing the current much sooner than the baseline, which once again agrees with the theoretical assumptions as before.

The simulations show the effects that the values of the individual elements have on the overall system, with the time constant being functions of various elements, usually in combination with each other. It is important to stress that these results are only applicable if the system under consideration is assumed to be ideal, since, in practice, it is nearly impossible to emulate electrical circuits without factoring in non-ideal behaviours in multiple aspects. These simulations provide a perspective into how the elements behave under the application of the two different types of input signals. While the analogue signal is a smooth ramp, the digital signal has several abrupt steps that force the sensor to behave differently, resulting in an output curve that, on the surface, appears to be identical but, when zoomed in, shows minor differences. The values of the circuit parameters also influence the step and ramp responses, as seen with the different coloured curves representing the responses of the different systems.

## 4. Conclusions

Analogue and digital modes of operation on the Metrohm VIONIC were used to test the sensor capabilities of a screen-printed electrode, with the working electrode modified with platinum on a carbon catalyst. The sensor element was used to detect hydroquinone in a hydrochloric acid medium, in the form of redox peaks while running CVs. Both analogue and digital CVs showed the peaks in identical positions, confirming the successful operation of the sensor element. The effect of the scan rate on the redox of hydroquinone was checked. It showed that the total measured currents were directly proportional to the scan rate in the case of both the analogue and digital CVs. While comparing individual analogue and digital scan rates, subtle differences were noticed between the two, with the analogue CVs producing smaller total currents compared to the digital CVs and some changes in the shape of the analogue curves when compared side by side with the digital ones. This proved that the type of potentiostat hardware does play a role in the results obtained in electrochemical measurements. Simulations confirmed the assumptions made about the nature of the input signals being responsible for the difference observed in the measured voltammograms. Step and ramp responses of the assumed ideal electrochemical systems showed that the individual elements of the electrochemical equivalent circuit determine how the applied signals, be it voltage or current, appear across the individual elements and how they are affected by the digital and analogue inputs. The results conclude that the use of analogue hardware vs. digital hardware causes some differences that are noticeable to the end user, though their effect on the underlying electrochemical processes may not be very significant.

## Figures and Tables

**Figure 1 sensors-24-04907-f001:**
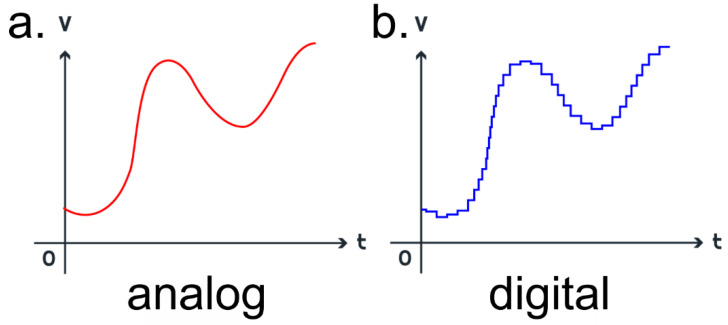
Exaggerated waveforms showcasing the infinitesimal differences between (**a**) an analogue signal and (**b**) a digital signal.

**Figure 2 sensors-24-04907-f002:**
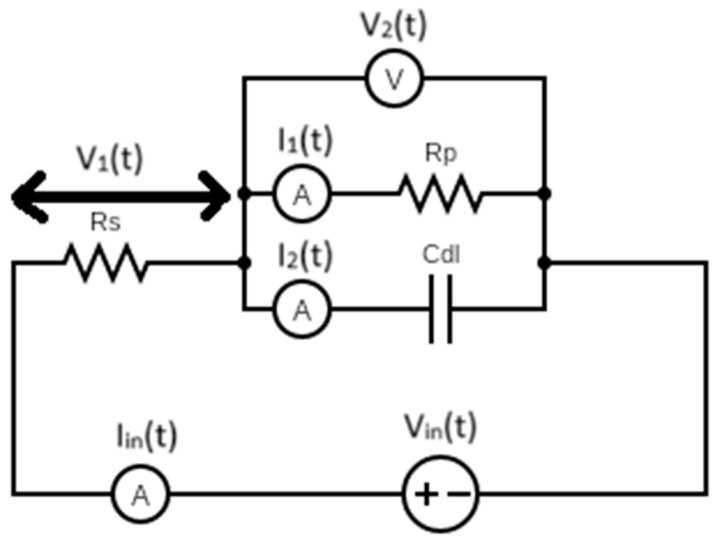
Randles’ circuit. Where, V_in_(t) = input voltage, I_in_(t) = input current, V_1_(t) = output voltage across the solution resistance R_S_, V_2_(t) = output voltage across the polarization resistance R_p_ and double-layer capacitance C_dl_, I_1_(t) = output current through the polarization resistance R_p_, I_2_(t) = output current through the double-layer capacitance C_dl._

**Figure 3 sensors-24-04907-f003:**
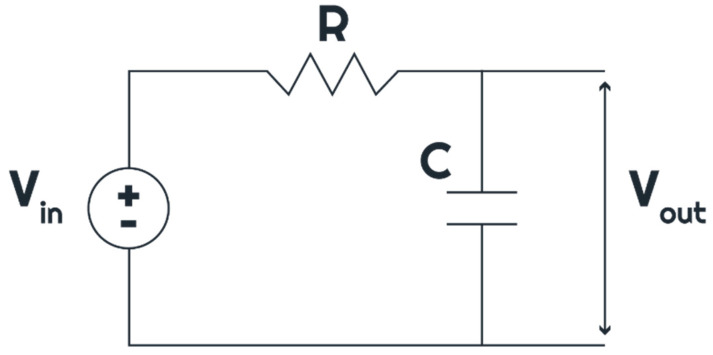
Series RC circuit.

**Figure 4 sensors-24-04907-f004:**
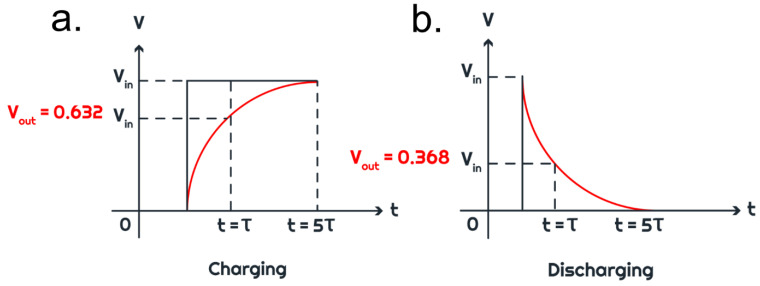
(**a**) Charging curve of series RC circuit. (**b**) Discharging curve of series RC circuit.

**Figure 5 sensors-24-04907-f005:**
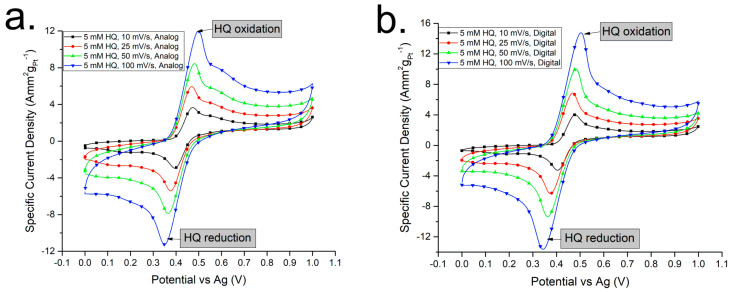
(**a**) Analogue CVs of hydroquinone redox at different scan rates. (**b**) Digital CVs of hydroquinone redox at different scan rates.

**Figure 6 sensors-24-04907-f006:**
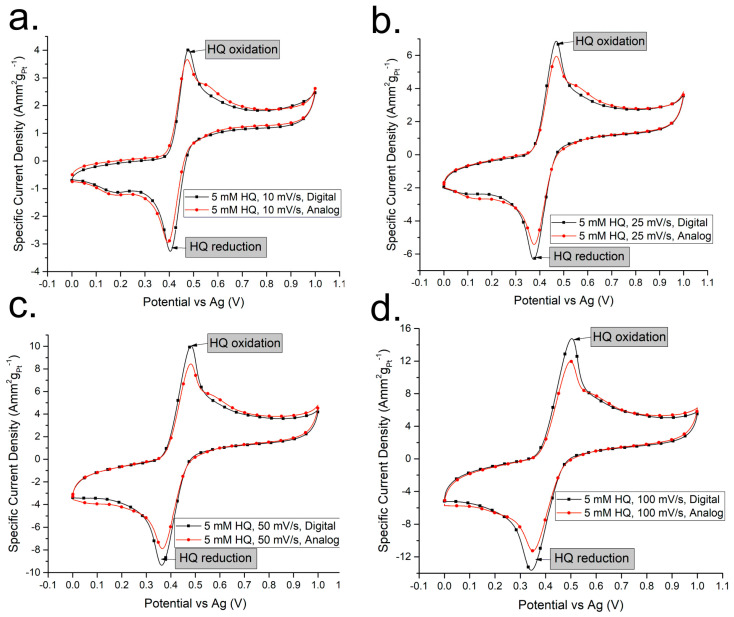
Analogue vs. digital CVs of hydroquinone redox at different scan rates: (**a**) 10 mVs^−1^, (**b**) 25 mVs^−1^, (**c**) 50 mVs^−1^, (**d**) 100 mVs^−1^.

**Figure 7 sensors-24-04907-f007:**
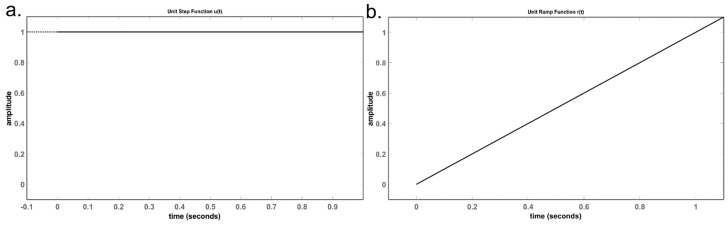
(**a**) Unit-step function. (**b**) Unit-ramp function.

**Figure 8 sensors-24-04907-f008:**
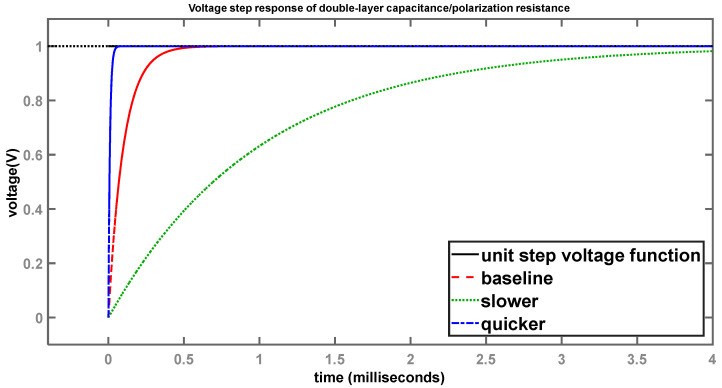
Voltage-step response of double-layer capacitance/polarization resistance.

**Figure 9 sensors-24-04907-f009:**
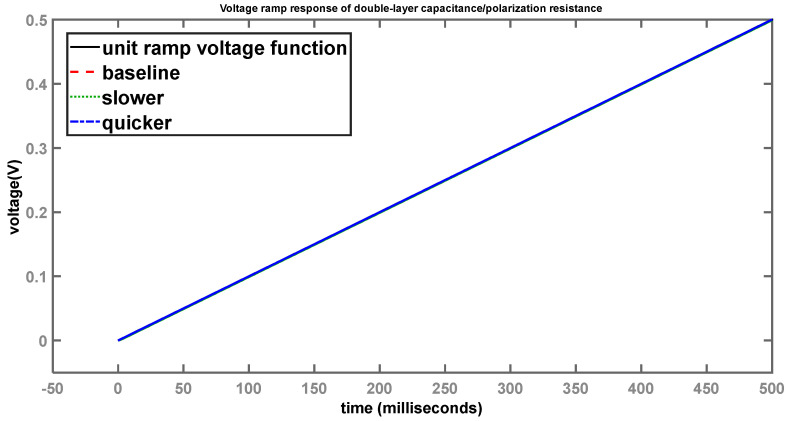
Voltage-ramp response of double-layer capacitance/polarization resistance (longer time scale).

**Figure 10 sensors-24-04907-f010:**
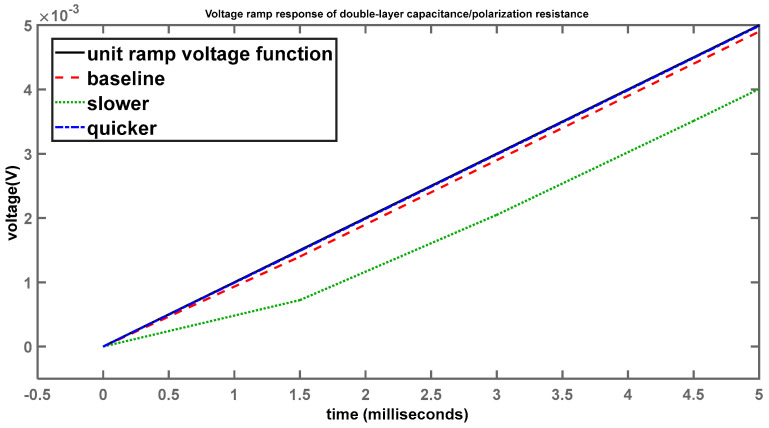
Voltage-ramp response of double-layer capacitance/polarization resistance (shorter time scale).

**Figure 11 sensors-24-04907-f011:**
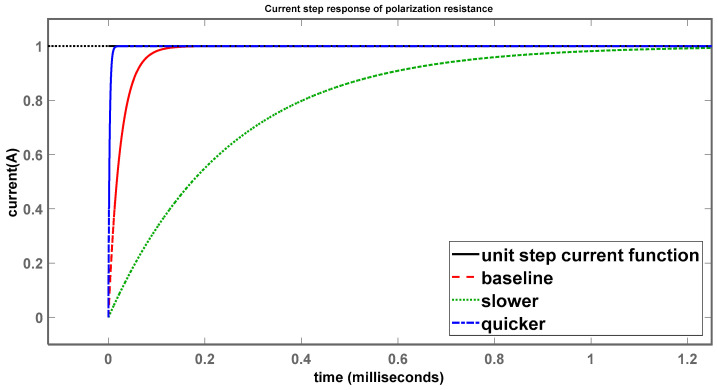
Current-step response of polarization resistance.

**Figure 12 sensors-24-04907-f012:**
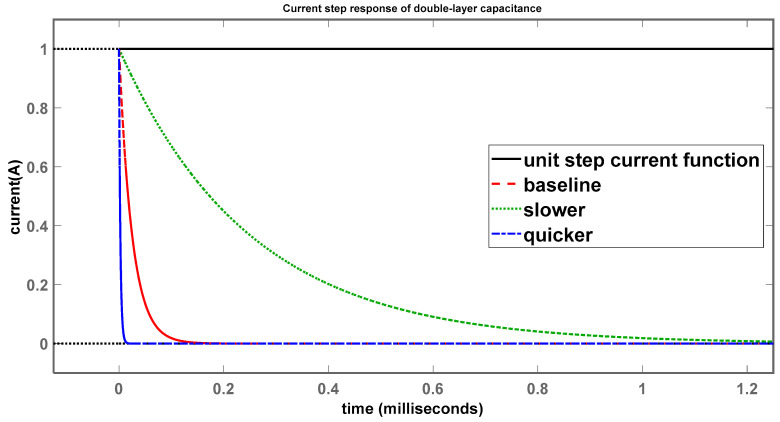
Current-step response of double-layer capacitance.

**Figure 13 sensors-24-04907-f013:**
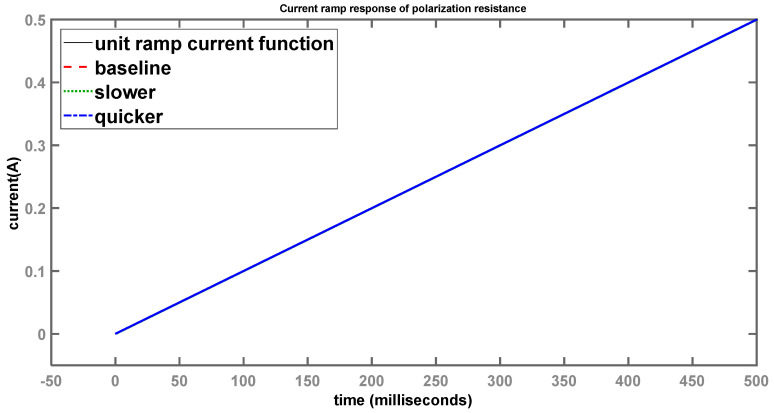
Current-ramp response of polarization resistance (longer time scale).

**Figure 14 sensors-24-04907-f014:**
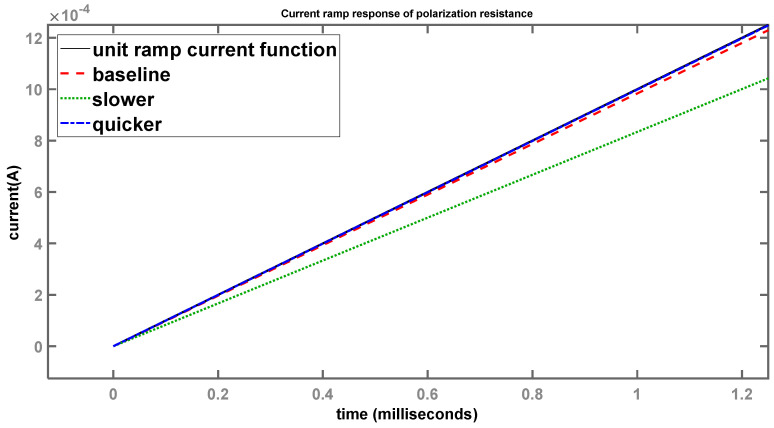
Current-ramp response of polarization resistance (shorter time scale).

**Figure 15 sensors-24-04907-f015:**
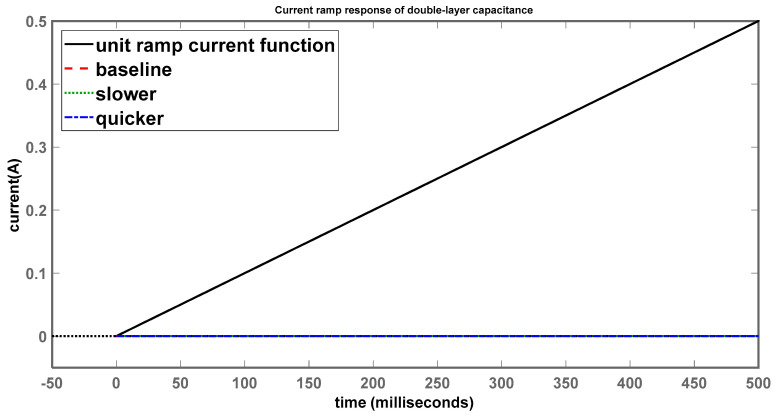
Current-ramp response of double-layer capacitance (longer time scale).

**Figure 16 sensors-24-04907-f016:**
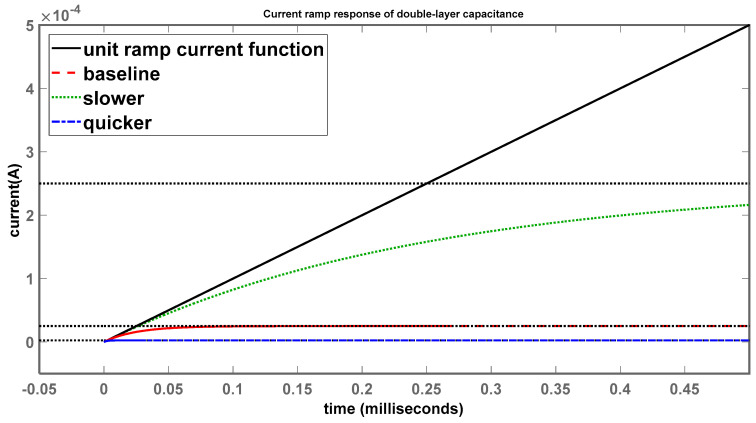
Current-ramp response of double-layer capacitance (shorter time scale).

## Data Availability

Data will be made available by the authors on request.

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
