# Peer review of "Influence of Potentiostat Hardware on Electrochemical Measurements"

_sensors, 2024, doi:10.3390/s24154907_

Round 1

Reviewer 1 Report

Comments and Suggestions for Authors

This paper analyzes the role of two operating modes of the same potentiostat, where  the redox processes of hydroquinone in hydrochloric acid media are analyzed. For this purpose, the behavior of cyclic voltammetry implemented by means of digital and analog scan is evaluated and compared.

The authors are suggested to focus on the following points:

1.- Abstract. It is too concise. The possible applications of the developments made and an anticipation of the results obtained are required.

2.- Introduction section. The novelties and contributions of this paper beyond the state of the art must be highlighted at the end of the introduction section, if necessary as a list of bullet points.

3.- There is no discussion about the influence of the internal parameters of the D/A converter, such as number of bits, rise time, etc. that determine how the quantization is done must play their role on the results. They could explain the differences seen in Figs. 5 and 6.

4.- Matlab simulations must include the quantization introduced by the digital system.

5.- Figs. 8-10 are illegible.

6.- The experimental section cannot be placed after the conclusions. It lacks of many details.

7.- References are old and scarce.

8.- In general the paper lacks of quality and needs a profound work in order it can be accepted for publications.

I hope my comments can help to enhance the quality of a future resubmission of a much worked version of the manuscript.

Comments on the Quality of English Language

I suggest proofreading the paper carefully.

Author Response

This paper analyzes the role of two operating modes of the same potentiostat, where  the redox processes of hydroquinone in hydrochloric acid media are analyzed. For this purpose, the behavior of cyclic voltammetry implemented by means of digital and analog scan is evaluated and compared.
We the authors thank the reviewer for taking the time to go through the manuscript and giving their valuable feedback on improving the submission for publication.

The authors are suggested to focus on the following points:

1.- Abstract. It is too concise. The possible applications of the developments made and an anticipation of the results obtained are required.
We thank the reviewer for this suggestion, the abstract has been expanded and some additional details have been added which are highlighted in yellow in the revised manuscript.

2.- Introduction section. The novelties and contributions of this paper beyond the state of the art must be highlighted at the end of the introduction section, if necessary as a list of bullet points.
We thank the reviewer for this suggestion, some additional discussion has been included about state of the art potentiostats available to end users, once again highlighted in yellow.

3.- There is no discussion about the influence of the internal parameters of the D/A converter, such as number of bits, rise time, etc. that determine how the quantization is done must play their role on the results. They could explain the differences seen in Figs. 5 and 6.
We thank the reviewer for bringing up this important point. Although we would like to take into consideration all the parameters at play, such as quantization errors and the total harmonic distortion introduced by the A-D and D-A conversions, unfortunately we don't have any information available about the internals of the potentiostat, nor do we know what algorithms are being employed within to address issues such as sampling. To our regret we have to treat the potentiostat as a black box as the end user, which is the manuscript has mainly focused on the electrochemical side of the discussion, since we have tools to accurately characterize that part of the equation and make reasonable assumptions which have been accepted by the electrochemistry community. To account for this omission, the following paragraph has been added (highlighted in yellow in the revision)
"While on the surface it’s easy to associate all the noticeable changes to the switching of the operational modes on the potentiostat, the internal electronics and the employed algorithms within the device also play a key role in how the data obtained is processed. Quantization errors caused by the conversion of the data between analog to digital form, which is necessary for the data to be processed by the computer can also be a factor, and the total harmonic distortion of the Analog-to-Digital Converter (ADC) determines the integrity of the signal after conversion, which is a function of the resolution of the ADC. However this information is not readily available to the end users and for all intents and purposes, the potentiostat is a black box for the end user.".

4.- Matlab simulations must include the quantization introduced by the digital system.
We thank the reviewer for this suggestion, while this was one of the main goals of the study, the lack of access to the information about the internal hardware makes it very difficult to even make theoretical assumptions about quantization errors and total harmonic distortion as mentioned in the previous point. Also as we are mainly dealing with electrochemistry, we are still learning on how to make much more complex simulations and decided to limit ourselves to easily available tools such as the step response within the MATLAB control system toolbox. To add more value to the manuscript, additional step and ramp response simulations for current along with voltage have been added. (highlighted in yellow in the manuscript).

5.- Figs. 8-10 are illegible.
We thank the reviewer for noticing the problems, and the figures have been addressed and made more legible with larger font sizes and thicker lines.

6.- The experimental section cannot be placed after the conclusions. It lacks of many details.
We thank the reviewer for pointing out this, the paragraph has been moved to follow the introduction.

7.- References are old and scarce.
We thank the reviewer for this comment, some newer references have been added with respect to electrochemical potentiostats and methods, although we would like to say that the older references used are from textbooks which have generally been more reliable than articles published about similar phenomenon.

8.- In general the paper lacks of quality and needs a profound work in order it can be accepted for publications.
We thank the reviewer once again for giving their valuable time and effort to improve the quality of the submission, and we hope that the revisions made meet the criteria to elevate our manuscript to a higher quality and potentially acceptance for publication.

I hope my comments can help to enhance the quality of a future resubmission of a much worked version of the manuscript.
We humbly thank the reviewer once again, and we hope our work is seen as an improvement in some regard, all the added changes have been highlighted in yellow to provide better clarity.

Reviewer 2 Report

Comments and Suggestions for Authors

The paper discusses the influence of hardware on electrochemical measurements but does not sufficiently elaborate on the significance of this issue and its specific implications for the field of electrochemistry. The authors are advised to further emphasize the importance of the research problem and its potential contribution to the advancement of electrochemical measurement techniques

1.Although the paper mentions simulations and experimental results, the specific contributions of this paper are not clearly described. The authors should clearly list the innovations and main contributions of this paper to quickly grasp the core value of the research.

2. The paper seems to lack a thorough discussion of the latest research advances, especially in the areas of electrochemical measurement and hardware technology. The authors are advised to supplement the current research trends in the field and how this paper is connected to or provides a new perspective on these advances.

3. The paper presents experimental results, but there seems to be insufficient in-depth analysis of the data. The authors are advised to provide more data analysis, including statistical tests or model fitting, to support the reliability of the research findings.

4.  The discussion section can be further expanded, including in-depth interpretation of the experimental results, comparative analysis with existing literature, and potential impact of the results on practical applications.

Comments on the Quality of English Language

it is ok

Author Response

The paper discusses the influence of hardware on electrochemical measurements but does not sufficiently elaborate on the significance of this issue and its specific implications for the field of electrochemistry. The authors are advised to further emphasize the importance of the research problem and its potential contribution to the advancement of electrochemical measurement techniques
We the authors thank the reviewer for providing their valuable time in order to improve the quality of our submission for publishing.

1.Although the paper mentions simulations and experimental results, the specific contributions of this paper are not clearly described. The authors should clearly list the innovations and main contributions of this paper to quickly grasp the core value of the research.
We thank the reviewer for the above suggestion, the main goal of the article was to find differences between the two operational modes of the potentiostat with respect to a common electrochemical method (cyclic voltammetry), which was successfully achieved. The abstract, the introduction and the conclusion have been modified to include the advances made with respect to potentiostat hardware (all changes have been highlighted in yellow).

2. The paper seems to lack a thorough discussion of the latest research advances, especially in the areas of electrochemical measurement and hardware technology. The authors are advised to supplement the current research trends in the field and how this paper is connected to or provides a new perspective on these advances.
We thank the reviewer for the above suggestion. While we tried to address this issue, unfortunately the lack of information about the nature of analog and digital hardware within the electrochemical community led us to not finding any comparable studies, and we accept this as not a big issue as electrochemists typically focus on the systems they are well versed with, such as batteries, fuel cells, sensors etc and generally treat the potentiostat as a black box for all intents and purposes. In addition, most potentiostat OEMs generally refrain from providing details about internal hardware and software to protect their IP. We have tried to utilize our knowledge to the best degree in order to understand the underlying effects of the applied signals, by breaking down the electrochemical system into an equivalent circuit, for which we have reliable methods and tools as opposed to focusing on what might be going on inside the potentiostat, which is equally important but behind locked doors.

3. The paper presents experimental results, but there seems to be insufficient in-depth analysis of the data. The authors are advised to provide more data analysis, including statistical tests or model fitting, to support the reliability of the research findings.
We thank the reviewer for the above suggestion. Some additional simulations have been added to compensate for the aforementioned lack of data, however with the electrochemical side of the story, the limited access to info with respect to the internal hardware of the potentiostat and the lack of knowledge on the implemented algorithms restricts our possibilities with respect to statistical tests, as we have access to just one method between which analog and digital modes can be differentiated. In the future, if there is a possibility that we have access to more electrochemical methods which can be performed with analog hardware, such as differential pulse voltammetry, AC voltammetry, Chronoamperometry and Impedance Spectroscopy, we would have a better sample size and data to dive deeper into statistical analysis.

4.  The discussion section can be further expanded, including in-depth interpretation of the experimental results, comparative analysis with existing literature, and potential impact of the results on practical applications.
We thank the reviewer for this suggestion, additional discussion has been included in the revision where we discuss the current step and ramp responses along with the previously included voltage step and ramp responses to go into further detail about how the individual elements within the equivalent electrochemical circuit play a role in the system's overall response to applied signals (highlighted in yellow in the revision)

Round 2

Reviewer 1 Report

Comments and Suggestions for Authors

The authors have improved the manuscript according to my comments but more references from 2023-2024 are still required

Comments on the Quality of English Language

The authors are suggested toproofread the manuscript

Author Response

The authors have improved the manuscript according to my comments but more references from 2023-2024 are still required

We the authors once again thank the reviewers for providing their time and energy to review the submission for the second time, and suggest more changes for the improvement of the final version of the manuscript. In accordance, new references regarding discussion about electrochemical potentiostats which were published since 2023 have been added, which have been typeset in blue to be easily distinguished from the previous round of modifications.

The authors are suggested toproofread the manuscript

We thank the reviewer for recommending the manuscript to be proofread, the final version was provided to native English speaker Dr. Paul McGuiness, who managed to proofread and improve the manuscript as quickly as possible which we hope will be satisfactory for the reviewers and editors. All the changes made by Dr. McGuiness have been shown in red, while the changes made by the authors are shown in blue for distinction.